# Large Estimate Variations in Assessed Energy Expenditure and Physical Activity Levels during Active Virtual Reality Gaming: A Short Report

**DOI:** 10.3390/ijerph20021548

**Published:** 2023-01-14

**Authors:** Jan-Michael Johansen, Kjartan van der Iest Schutte, Solfrid Bratland-Sanda

**Affiliations:** Department of Sports, Physical Education and Outdoor Studies, University of South-Eastern Norway, 3800 Bø, Norway

**Keywords:** physical activity, exergaming, virtual training, energy expenditure

## Abstract

The purpose of the study was to compare methods for estimating energy expenditure (EE) and physical activity (PA) intensity during a 30 min session of active virtual reality (VR) gaming. Eight individuals (age = 25.4 ± 2.0 yrs) participated, with a maximal oxygen consumption (VO_2max_) of 41.3 ± 5.7 mL∙kg^−1^∙min^−1^. All tests were conducted over two days. An incremental test to determine the VO_2max_ when running was performed on day 1, while 30 min of active VR gaming was performed on day 2. The instruments used for EE estimations and PA measurements were indirect calorimetry, a heart rate (HR) monitor, and waist- and wrist-worn accelerometer. Compared to indirect calorimetry, waist-worn accelerometers underestimated EE (mean difference: −157.3 ± 55.9 kcal, *p* < 0.01) and PA levels. HR-based equations overestimated EE (mean difference: 114.8 ± 39.0 kcal, *p* < 0.01 and mean difference: 141.0 ± 81.6 kcal, *p* < 0.01). The wrist-worn accelerometer was the most accurate in estimating EE (mean difference: 23.9 ± 45.4 kcal, *p* = 0.95). The large variations in EE have implications for population-based surveillance of PA levels and for clinical studies using active VR gaming.

## 1. Introduction

Online and virtual training are trending worldwide [1]. One emerging concept is active video gaming, also referred to as exergaming, with increased use of virtual reality (VR) systems [2]. Active VR-based training has been launched as a useful and enjoyable activity for both the general population and for clinical populations [3,4,5,6].

Active video games increase light to moderate-intensity physical activity (PA) in children and adolescents [7,8], who have similar acute responses in terms of their heart rate (HR), oxygen uptake (VO_2_), and energy expenditure (EE) [9]. EE has been found to be comparable to traditional activities, e.g., walking and running [10]. However, previous studies have reported that different VR games, game difficulty levels, and VR equipment all have an impact on both EE and the PA level [3,10,11].

World Health Organization (WHO) updated its guidelines on PA and sedentary behavior in 2020 and recommend either 150–300 min/week of moderate-intensity PA, 75–150 min/week of vigorous-intensity PA, or a combination of the two for adults [12]. Furthermore, the updated guidelines highlight that instead of the previous recommendations of a minimum of 10 min bouts of PA for it to count, all PA and every move now counts when accumulating daily PA. To limit sedentary behavior is now also a separate recommendation in the 2020 guidelines. This means that PA of very light to light intensity may also be of utmost importance to mitigate the consequences of sedentary behavior. Troiano et al. [13] state that the updated guidelines from the WHO impact global PA monitoring. Such surveillance has been conducted mainly through self-reporting, although device-based measures are recommended [14]. However, well-known limitations of portable devices such as accelerometers and heart rate (HR) monitors call under question the accuracy of such devices for determining EE during active VR gaming [3,10,15]. For instance, accelerometers attached to the wrist and waist have previously shown large differences in estimations of both sedentary activity and moderate-to-vigorous physical activity (MVPA) during active VR gaming [3]. In that study, waist-worn accelerometers seemed, consequently, to estimate a lower exercise intensity compared to wrist-worn accelerometers during a variety of active VR games. However, both tools provided similar estimates of the exercise intensity while walking. This may indicate an underestimation of EE by waist-worn accelerometers when the upper body is predominantly used, or an overestimation by wrist-worn accelerometers in active VR games. Perrin et al. [10] criticize the use of the heart rate (HR) to evaluate intensity and EE during active video gaming, and reported a possible overestimation of exercise intensity, and thus EE, by HR measures. Pope et al. [15] reported less valid estimations of EE by well-known HR-based smartwatches when compared to accelerometers, although without comparisons with indirect calorimetry.

Accurate methods to measure the PA level and EE during a variety of physical activities are crucial in both practical and clinical settings and in science. Self-tracking of EE has also been reported to be positively associated with better body composition and health indicators [16]. Yet, the most accurate methods for the assessment of EE, i.e., indirect calorimetry and VO_2_ uptake, are resource demanding and not applicable in practical, everyday life settings. More accessible methods such as HR-based activity trackers and wearable devices must ensure their accuracy across both traditional activities such as walking, running, and strength training and new and emerging activities such as active VR gaming. Their success in this regard is uncertain as, to date, the scientific field of EE in active video and VR gaming is lacking comparisons of wearable devices to indirect calorimetry.

Therefore, the aim of this study was to compare different measurements of EE with indirect calorimetry during active VR gaming. Our hypotheses were: (1) wrist-worn accelerometers would estimate EE more accurately than waist-worn accelerometers, and (2) HR measurements would overestimate EE.

## 2. Materials and Methods

### 2.1. Participants

Eight healthy individuals (6 males and 2 females) participated in this study. The participants were recruited from advertisements in the local community (i.e., university campuses and health service offices) and through social media. All participants were young adults and had a sedentary lifestyle with a self-reported PA level below the WHO guidelines. There were no self-reported lifestyle-related diseases among the participants. Written informed consent was provided by all participants after receiving information about the study and their option to withdraw from the study without further reason. The study was conducted in accordance with the Declaration of Helsinki. The study protocol was evaluated by the regional ethics committee (REC, 181005) and approved by the Norwegian Centre for Research Data (NSD, 140188).

### 2.2. Test Procedures

On the first visit to the laboratory, the participants performed an incremental test to determine their VO_2max_ when running. Prior to the incremental protocol, all participants performed a self-selected warm-up for 10 min. The starting intensity was set to correspond to approximately 70% of the participants’ predicted VO_2max_. This intensity was based on the participants’ self-reported PA levels and evaluations from experienced research personnel. The treadmill velocity was increased every 30 s by 0.5 km∙h^−1^ until voluntary exhaustion. The inclination was held at 5% throughout the whole test. Voluntary exhaustion, flattening of the VO_2_ curve, and a RER > 1.10 were used to evaluate if the VO_2max_ was reached. The mean of the two highest consecutive 20 s VO_2_ measurements was used to determine the VO_2max_. The peak heart rate (HR_peak_) was determined by the highest HR achieved during the test.

Sometime 2–7 days after the VO_2max_ test, a 40 min VR-testing session was performed in the laboratory. To avoid any possible impact of the incremental VO_2max_ test on the EE and overall performance in the active VR gaming session, a minimum of two days between the test and the gaming session was required. The variation beyond two days apart was due to practical reasons. Participants were instructed to live as they normally would between test days to avoid any confounding factors affecting the EE calculations. The testing session consisted of a 10 min warm-up followed by 30 min of active VR gaming. The warm-up served as a familiarization with the game and VR equipment. During the active VR gaming session, participants were allowed to self-select their preferred playing level. To ensure continuous gameplay, the participants used a “no fail” version of the game. The HR was registered every 30 s, and the average HR (HR_ave_) was calculated as the mean of all HR measurements. The average VO_2_ (VO_2ave_) was calculated as the mean of all 20 s VO_2_ measurements during the 30 min active VR gaming session. All participants wore two accelerometers during the gaming session: one attached to the right hip, and one attached to the dominant wrist.

### 2.3. Measurements

VO_2_ measurements in the VO_2max_ test and active VR gaming session were performed by a Jeager Vyntus CPX (CareFusion, GmbH, Hoechberg, Germany) with a mixing chamber every 20 s. Before testing, the gas analyzers were calibrated with ambient air and certified calibration gases (16.00% O_2_/5.00% CO_2_). The flow sensor was calibrated by standard flow volumes of 2.0 L and 0.2 L. The incremental running test was conducted on a Woodway PPS 55 Sport (Waukesha, WI, USA). The HR was registered by Polar s610 HR monitors (Kempele, Finland) throughout all tests.

The VR game used in this study was “Beat Saber” (Beat Games, Prague, Czech Republic), which is a rhythmic game where cubes and walls are coming toward the player at a certain velocity and to the beat of a song. The aim of the game is to slash as many cubes and dodge as many walls as possible. The game consists of five levels of difficulty (Easy–Expert+), where the numbers and velocity of cubes and walls increase at higher difficulty levels. An overview of the total numbers of cube slashes and wall dodges in the different difficulty levels of the game is presented quantitatively in Table 1. The VR equipment used was an Oculus Quest headset (Oculus, Meta Platforms Inc., Menlo Park, CA, USA), with two handheld controls. Activity counts, sedentary behavior, and MVPA were measured by two ActiGraph GT9X accelerometers (Pensacola, FL, USA) during the gaming session. Activity counts were captured in 1 s epoch intervals [3]. Accelerometers were initialized to start recording at the start of the active VR gaming session and to stop recording after 30 min. For initializing, downloading, and analyses, we used the ActiLife software program (ActiGraph, LLC, Pensacola, FL, USA). Freedson cut-off points [17] were used to define sedentary activity, light PA (LPA), and MVPA.

### 2.4. Energy Expenditure Estimations

The EE was calculated using indirect calorimetry in a standard manner based on the table of non-protein respiratory quotient developed by Zuntz and colleagues and presented by McArdle et al. [18]. To estimate the EE from HR measurements, the two equations (HR_eq.1_ and HR_eq.2_) presented in Keytel et al. [19] were used. These equations were as follows:HR_eq.1_: EE = [−59.3954 + gender × (−36.3781 + 0.271 × age + 0.394 × body weight + 0.404 × VO_2max_ + 0.634 × HR_ave_) + (1 − gender) × (0.274 × age + 0.103 × body weight + 0.380 × VO_2max_ + 0.450 × HR_ave_)] × duration in minutes/4.184,
and
HR_eq.2_: EE = gender × (−55.0969 + 0.6309 × HR_ave_ + 0.1988 × body weight + 0.2017 × age) + (1 − gender) × (−20.4022 + 0.4472 × HR_ave_ − 0.1263 × body weight + 0.074 × age) × duration in minutes/4.184.

Since these equations estimate the EE in kilo joule (kJ) per minute, the product was divided by 4.184 for conversion to kilocalories [18]. To express the total EE in the session, the product of the equations was multiplied by 30 min.

For EE estimation from the accelerometers, the ActiLife software was used. For the wrist-worn accelerometer, the wrist-worn mark was checked as recommended by the manufacturer [20].

### 2.5. Statistics

Statistical analysis was performed by the Statistical Package for Social Science (SPSS) version 28, and the level of significance was set to an alpha level of *p* < 0.05. QQ plots and normality tests indicated a normal distribution, and descriptive statistics (mean ± SD) were used. A univariate GLM with Tukey’s post hoc test, intraclass correlation coefficients (ICC), and Bland–Altman plots was used to analyze the differences between EE estimation methods. Paired sample t-tests were used to evaluate differences in intensity and EE between waist- and wrist-worn accelerometers.

## 3. Results

All participants adhered to the protocol. Age, body weight, and physiological results from the VO_2max_ test and the 30 min active VR gaming session (i.e., PA levels and EE from indirect calorimetry, HR, and accelerometers) are presented in Table 2.

PA characteristics from the active VR gaming session are presented in Table 2. During the VR session, no participants played at the lowest level of difficulty (Easy, level 1), with the average level of difficulty of 3.5 indicating that the most-played difficulty levels were Hard (3) and Expert (4). The MET rate was 3.8 ± 1.0, and the relative intensity was 31.6 ± 5.6% of VO_2max_ and 60.6 ± 6.3% of HR_peak_. The waist- and wrist-worn accelerometers displayed 431.1 ± 202.5 and 18 265.7 ± 5616.3 CPM, respectively. In terms of PA intensity, the sedentary time was 22.6 ± 6.5 vs. 1.2 ± 0.7, LPA was 4.9 ± 6.1 vs. 1.8 ± 0.8, and MVPA was 2.5 ± 0.7 vs. 27.0 ± 1.3 for the waist- and wrist-worn accelerometers, respectively. Significant differences in CPM, sedentary time, MVPA, and total EE were observed between wrist- and waist-worn accelerometers (all *p* < 0.01).

No difference in EE was observed between indirect calorimetry and the wrist-worn accelerometer (MD: 23.9 ± 45.4 kcal, 95% CI = 120.1, −72.2, *p* = 0.95). The mean differences between HR equations and indirect calorimetry were 114.8 ± 39.0 kcal (95% CI = 226.2, 27.1, *p* < 0.01) and 141.0 ± 81.6 kcal (95% CI = 240.5, 41.5, *p* < 0.01) for HR_eq.1_ and HR_eq.2_, respectively. For the waist-worn accelerometer, the MD from indirect calorimetry was −157.3 ± 55.9 kcal (95% CI = −61.2, −253.5, *p* < 0.01). Accordingly, Bland–Altman plots (Figure 1A–D) illustrate an overestimation of EE by the HR and an underestimation of EE by the waist-worn accelerometer. The ICC (95% confidence interval) between indirect calorimetry and HR_eq.1_, HR_eq.2_, and waist- and wrist-worn accelerometers were 0.59 (−0.11, 0.93), 0.52 (−0.05, 0.90), 0.06 (−0.09–0.43), and 0.81 (0.20, 0.96), respectively.

## 4. Discussion

The main findings were that EE was overestimated by HR and underestimated by waist-worn accelerometers, while wrist-worn accelerometers provided a more accurate estimation of EE compared to indirect calorimetry.

EE was largely underestimated by waist-worn accelerometers compared to indirect calorimetry, and the finding that they give low PA intensity readings compared to actual values is consistent with the findings of Evans et al. [3]. This underestimation and the apparently low quantity of kilocalories expended may be due to the nature of the active VR game. Upper-body movements dominate the game at the easy-to-medium levels, and movements that are more whole-body oriented, such as dodging incoming walls, are of minor importance until the highest level of difficulty (Table 1). This may explain why wrist-worn accelerometers estimated EE more accurately.

There is a lack of studies estimating EE using both waist- and wrist-worn accelerometers compared to EE calculated by indirect calorimetry in active VR games; therefore, this study provides novel and important data on the evaluation of EE using wearable devices during an active VR game. In addition, the extensive use of waist-worn accelerometers to track daily PA in larger population studies [13] may potentially lead researchers to miss valuable information on stationary PA concentrated mainly in the upper body. This is a concern for scientific studies evaluating PA monitoring in the population.

The present study revealed that the EE during the 30 min active VR gaming session, when estimated by HR equations, was largely overestimated (by 78.1%) when using HR_eq.2_ compared to indirect calorimetry. When controlling for the individual VO_2max_ (HR_eq.1_), the overestimation was reduced to 63.6%. The large overestimation of EE by HR measurements in active gaming is in accordance with previous findings [10,15,21]. It is widely accepted that HR can be influenced by both psychological and physiological factors [10,22]. Increased and high HRs have been reported in 2D games with high cognitive strain [23], and in comparisons of active video games with walking or running, without corresponding differences in EE [10,21]. Therefore, the overestimation of EE may be a result of the sympathetic stress responses to the game and/or the VR equipment itself, and not the actual bodily movement. Although Pope et al. [15] reported somewhat better estimations of EE by HR monitors compared to the present study, that study did not compare EE estimations with indirect calorimetry. With the increased use of active or sedentary VR gaming [2], overestimation might influence the determination of EE assessed by instruments that include HR monitoring. The increased availability of activity trackers and HR monitors for the general population [24], combined with active VR gaming trending worldwide [1], may lead to invalid data on EE, PA, and training intensity. Since HR monitors are widely used to evaluate and track EE, exercise, and health indicators [16,24], the development of more accurate equations for estimating EE from HR measurements in active VR gaming is urgently needed. Therefore, we argue that HR measurements should not be used, either in research or in clinical practice, as the only indicator and evaluator of PA intensity or EE during active VR gaming sessions.

The findings on EE above sedentary levels confirm the findings of previous investigations on active VR or video gaming [3,4,5,6,8,10,15]. The MET rate in the present study corresponds to very light to light PA, while the HR_ave_ corresponds to light-to-moderate PA [25]. Consequently, the game used in the present study seems to generate a low amount of PA, and thus, it seems to be insufficient to support the WHO guidelines on increased exercise at moderate or vigorous intensities in adult populations. However, the updated guidelines highlight that every move counts, and that it is worthwhile to limit sedentary behavior. This means that PA of very light to light intensity may also be of importance to mitigate the consequences of sedentary behavior and to create a more physically active adult population. Furthermore, active video and VR games have shown to have high scores for enjoyment and to be a motivating form of PA [3,6].

The significant differences between waist- and wrist-worn accelerometers when measuring PA intensity are in line with a previous study of the same game [3]. These findings may have implications for PA monitoring studies. It is important to capture such activities accurately as the revised PA recommendations from the WHO highlight that “every move counts” [12]. Furthermore, the determination of EE and PA intensity can be important in clinical studies where these variables are central to the effects on, e.g., treatment outcomes. The reduced accuracy of EE estimated from waist-worn accelerometers and HR equations requires particular attention.

The present study was conducted in accordance with recent recommendations on validation studies of estimated EE by wearable devices and smartphones [26]. In line with such recommendations, our calculation of the EE from indirect calorimetry in controlled laboratory conditions is a major strength of the study. In addition, the simultaneous estimations of EE by VO_2_, HR, and waist- and wrist-worn accelerometers is novel since this was lacking in previous investigations of active VR games. However, the low number of participants, homogenous sample of sedentary young adults, and lack of evaluation of several active VR games limit the generalization of the findings, meaning the present data should be interpreted with caution in relation to other populations. Nevertheless, the present study creates a foundation for future studies of larger cohorts and with greater heterogeneity in both samples and types of active VR games. In an additional limitation, we cannot rule out the possibility of differences in habitual lifestyle, such as the sleeping pattern or food intake, prior to the two test days. Yet, all participants were instructed to live as normal before the test days; therefore, we consider such factors to have had a minimal impact on the present results. As a final point to note, one outlier was observed in the dataset, which may have influenced the results, but analyses without this participant revealed the same results as presented. Therefore, this participant was included in the final analyses and results.

## 5. Conclusions

To conclude, a 30 min active VR gaming session identified that wrist-worn accelerometers generated more accurate EE estimates compared to indirect calorimetry, while HR and waist-worn accelerometers showed inaccurate estimates. The findings may have methodological implications for population-based surveillance studies of PA levels due to increased participation in active VR gaming. However, these conclusions are based on one type of active VR game in a homogenous population of young, sedentary adults. Thus, future studies should investigate these differences in a broader range of active VR games (for instance, games that force the player to move physically over a smaller area, and games which incorporate both upper- and lower-body movements), and they should include other populations. 

## Figures and Tables

**Figure 1 ijerph-20-01548-f001:**
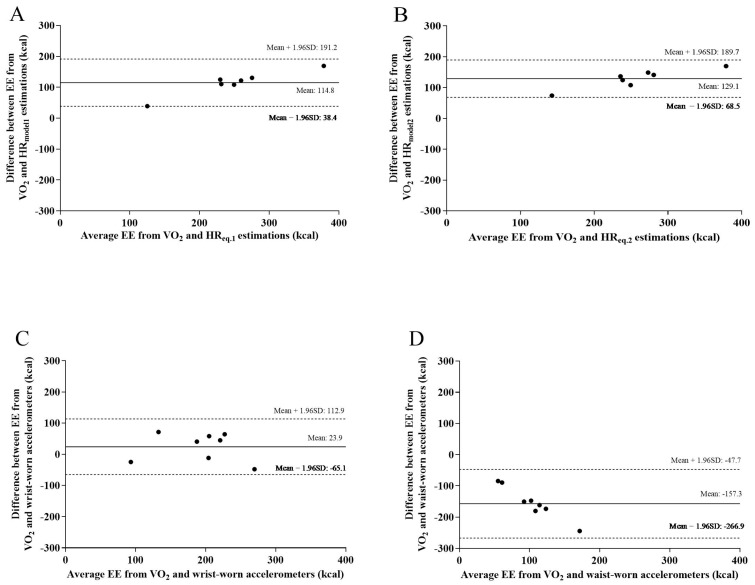
Bland–Altman plots: Differences in average values of energy expenditure (EE) from oxygen consumption (VO_2_) and either heart rate (HR) models (**A**,**B**) or accelerometers (**C**,**D**). Mean difference and 95% limits of agreement (mean ± 1.96 SD) are also displayed.

**Table 1 ijerph-20-01548-t001:** Descriptive data from the various levels of difficulty in the active VR game “Beat Saber”.

Levels of Difficulty	Slashes				Walls			
	Total	Right Hand	Left Hand	Slashes per Second	Total	Upper	Right	Left
1 = Easy	327	174	153	1.5	28	11	10	7
2 = Normal	472	237	235	2.2	22	11	7	4
3 = Hard	706	386	320	3.4	27	10	10	7
4 = Expert	926	505	421	4.4	27	10	10	7
5 = Expert+	1026	590	536	4.9	45	11	17	17

Values are the total number of cubes to slash with both hands, the right hand, and the left hand, and the slashes per second that the player should try to perform at the different difficulty levels. The total number of walls, upper walls, right walls, and left walls that the player should try to dodge at the different levels of difficulty are also displayed. The values indicate the number of slashes and walls that appear throughout the song each time the same song is played.

**Table 2 ijerph-20-01548-t002:** Physiological and active VR gaming characteristics at an individual and a group level.

	Subject Characteristics				Active VR Gaming Characteristics								
					PA Characteristics				Energy Expenditure (kcal)				
Subjects	Age (yr)	BW (kg)	VO_2max_ (mL∙kg^−1^∙min^−1^)	HR_peak_ (b∙min^−1^)	VO_2ave_ (mL∙kg^−1^∙min^−1^)	HR_ave_ (b∙min^−1^)	MVPA (Waist)	MVPA (Wrist)	VO_2_	HR_eq.1_	HR_eq.2_	Waist	Wrist
Individual level													
1 (F)	27	86.2	29.6	-	7.8	-	1.7	25.4	97.3	-	-	13.8	168.1
2 (F)	22	60.8	41.3	202	12.0	114.6	2.7	26.1	105.5	144.5	179.8	14.1	80.8
3 (M)	24	99.4	46.2	204	20.0	151.4	3.2	29.0	293.8	462.6	463.8	32.4	245.6
4 (M)	26	91.2	47.4	212	15.9	128.0	3.6	27.0	210.0	340.6	351.3	29.7	198.2
5 (M)	27	100.5	37.3	201	13.4	123.7	1.9	28.3	198.7	320.2	347.5	18.1	242.8
6 (M)	24	102.6	41.3	199	11.9	113.7	2.5	26.7	176.1	286.4	300.7	28.4	233.6
7 (M)	25	106.4	43.5	200	12.5	113.3	2.6	27.7	195.2	303.7	303.4	32.5	258.9
8 (M)	28	97.1	43.8	201	12.0	115.1	1.9	25.9	167.4	292.0	303.9	16.5	207.5
Group level													
Mean	25.4	93.0	41.3	203	13.2	123	2.5	27.0 **	180.5	307.2 *	321.5 *	23.2 *	204.4
±SD	2.0	14.5	5.7	4	3.5	14	0.7	1.3	62.1	93.6	84.6	8.3	58.1

Values are individual values, means, and standard deviations. Yr, years. BW, body weight. VO_2max_, maximal oxygen consumption. ml kg^−1^ min^−1^, milliliters per kilogram body weight per minute. HR_peak_, peak heart rate. VO_2ave_, average oxygen consumption. HR_ave_, average heart rate. MVPA, moderate-to-vigorous physical activity in minutes. Waist, waist-worn accelerometer. Wrist, wrist-worn accelerometer. Kcal, kilocalories. HR_eq.1_, heart rate equation 1 from Keytel et al. [19]. HR_eq.2_, heart rate equation 2 from Keytel et al. [19]. F, female. M, male. * *p* < 0.01 significantly different from indirect calorimetry (VO_2)._ ** *p* < 0.01 significantly different from waist-worn accelerometer.

## Data Availability

The data that support the findings of this study are available from the corresponding author on reasonable request.

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
