# Peer review of "Large Estimate Variations in Assessed Energy Expenditure and Physical Activity Levels during Active Virtual Reality Gaming: A Short Report"

_ijerph, 2023, doi:10.3390/ijerph20021548_

Round 1
Reviewer 1 Report
Abstract could use a little grammar work with VO2 area not really being sentences.
Intro Section:
- Good info on lack of research out there about VR specifically. I’d like a little bit more background showing consistency or lack thereof for EE prediction with indirect calorimetry, VO2, wearables, and HR measurements.
Methods:
- I’d like to see a quick statement about how subjects were recruited: convenience, fliers at a gym, etc.
- Also, should have a little bit of info on physical activity level of subjects prior to study. Were they aerobically fit, regular exercisers, or sedentary? What kind of co-morbidities did they have? For generalization across populations, I’d want to know if these subjects were fit because if I then go to use the VR on a sedentary person, will I get the same EE prediction?
- How did you determine subject’s predicted VO2 max? Include that here.
Results
- No real values given to show subjects were homogeneous in their performance. I see some variance in their performances in Table 2, so you need to say how you ensured they were all similar enough that their performance variance didn't impact EE measurement results. Meaning, were there any large outliers for subject’s that could impact any of these calculations since the N is so small?
- You didn’t discuss at all why there were differences in time from treadmill to VR assessments. I’d like to see a bit more info in methods as to why there was such variance in time between tests and then in the results section, did the time between TM and VR change outcomes 2-7 days? I would suspect it wouldn’t make a difference, but you didn’t double check that and I’d want to know that to make sure it wasn’t a compounding factor in your results.
Discussion
- Good job with discussion of why waist monitors may have performed poorly and the HR increase due to stress responses to the game.
- You did do a slight comment on your subjects’ average MET levels only being 3.8. That is not that high for such a young population. I’d like to see a short comparison that to other activities (perhaps running since you tested that). Maybe expand that paragraph a bit more with how this could really impact WHO guidelines when it is very light to light PA. How could it really be used to increase physical activity minutes?
Overall, I like this paper and that you are exploring VR gaming as a means to encourage physical activity. Some minor edits would improve this.
Author Response
First of all, thank you for a thorough review of our manuscript. Please see our point-by-point response to your comments below.
1. Abstract could use a little grammar work with VO2 area not really being sentences.
Author response:
Thank you for pointing out an incomplete sentence in the abstract, this has now been revised accordingly
2. Intro Section:
Good info on lack of research out there about VR specifically. I’d like a little bit more background showing consistency or lack thereof for EE prediction with indirect calorimetry, VO2, wearables, and HR measurements.
Author response:
We have now revised the following paragraph (Line 49) and we hope that this gives a better overview of the background: “However, well-known limitations of portable devices such as accelerometers and heart rate (HR) monitors question the accuracy of such devices for determining EE during active VR gaming [3,10,15]. For instance, accelerometers attached to the wrist and waist have previously shown large differences in estimations of both sedentary activity and moderate-to-vigorous physical activity (MVPA) while active VR gaming [3]. In that study, waist-worn accelerometers seemed to consequently estimate a lower exercise intensity compared to wrist-worn accelerometers over a variety of active VR games. However, both sites provided similar estimates of exercise intensity during walking. This may indicate an underestimation of EE by waist-worn accelerometers when the upper-body is predominantly used, or an overestimation by wrist-worn accelerometers in active VR games. Perrin et al. [10] criticizes the use of heart rate (HR) to evaluate intensity and EE during active video gaming, and reported a possible overestimation of exercise intensity, and thus EE, by HR-measurements. Pope et al. [15] reported less valid estimations of EE in comparisons of accelerometers and well-known HR based smartwatches, although without comparisons to indirect calorimetry.”
3. Methods:
I’d like to see a quick statement about how subjects were recruited: convenience, fliers at a gym, etc.
Author response:
We have now included the following in the method-section in the revised manuscript (Line 82): “The participants were recruited from advertisements in the local community (i.e., university campuses and health service offices) and through social media.”
4. Also, should have a little bit of info on physical activity level of subjects prior to study. Were they aerobically fit, regular exercisers, or sedentary? What kind of co-morbidities did they have? For generalization across populations, I’d want to know if these subjects were fit because if I then go to use the VR on a sedentary person, will I get the same EE prediction?
Author response:
We have now included the following in the method-section in the revised manuscript (Line 84): “All participants were young adults and had a sedentary lifestyle with a self-reported PA level below the WHO guidelines. There were no self-reported lifestyle-related diseases among the participants.”
We have also revised this sentence in the discussion-section (Line 336): “However, the low number of participants, a homogenous sample of sedentary young adults and lack of evaluations of several active VR games limit generalization and the present data should be interpreted with caution in relation to other populations.”
5. How did you determine subject’s predicted VO2 max? Include that here.
Author response:
We have now revised this sentence to (Line 101): “The starting intensity was set to correspond to approximately 70% of the participants predicted VO2max. This intensity was based on the participants self-reported PA levels and evaluations from experienced research personnel.”
6. Results
No real values given to show subjects were homogeneous in their performance. I see some variance in their performances in Table 2, so you need to say how you ensured they were all similar enough that their performance variance didn't impact EE measurement results. Meaning, were there any large outliers for subject’s that could impact any of these calculations since the N is so small?
Author response:
One possible outlier was observed in the dataset, and we have now performed the statistical analyzes with and without this participant. The results show similar results with and without this participant included. We have included this sentence in the discussion part (Line 344): “One outlier was observed in the dataset that may have influenced the results but analyzes without this participant revealed the same results as presented. Therefore, this participant was included in the final analyzes and results.”
7. You didn’t discuss at all why there were differences in time from treadmill to VR assessments. I’d like to see a bit more info in methods as to why there was such variance in time between tests and then in the results section, did the time between TM and VR change outcomes 2-7 days? I would suspect it wouldn’t make a difference, but you didn’t double check that and I’d want to know that to make sure it wasn’t a compounding factor in your results.
Author response:
Thank you for your comment. The time between test days were chosen to avoid any possible impact of soreness or fatigue from the incremental VO2max test. We considered it to be sufficient with a minimum of two days between tests to avoid these confounding factors. The actual number of days between test days for each participant was mainly determined by practical reasons. However, we have now included this in the methods-section (Line 112): “To avoid any physiological impact of the incremental VO2max test on EE and overall performance in the active VR gaming session, a minimum of two days between the test and the gaming session was required. The variation beyond two days apart were due to practical reasons. Participants were also instructed to live as they normally did between test days to avoid any confounding factors for the EE calculations.”
We have also included this sentence in the discussion (Line 341): “Additionally, we cannot rule out the possibility of differences in habitual lifestyle, such as sleeping pattern or food intake, prior to the two test days. All participants were in-structed to live as normal between test days, and we therefore consider such factors to have a minimal impact on the present results.”
8. Discussion
Good job with discussion of why waist monitors may have performed poorly and the HR increase due to stress responses to the game.
You did do a slight comment on your subjects’ average MET levels only being 3.8. That is not that high for such a young population. I’d like to see a short comparison that to other activities (perhaps running since you tested that). Maybe expand that paragraph a bit more with how this could really impact WHO guidelines when it is very light to light PA. How could it really be used to increase physical activity minutes?
Author response:
Thank you for your comment, and we do see your point here. We did a test in running, but this was an incremental test. Since the protocol used for the running test did not provide a VO2 steady-state condition, unfortunately no accurate EE calculation or MET evaluation can be performed. Therefore, we are not able to compare the present MET results from the active VR gaming session to other activities directly. However, we have included the following in the discussion part (Line 312): “The findings of EE above sedentary levels confirm findings in previous investigations on active VR or video gaming [3-6,8,10,15]. The MET rate in the present study corresponds to very light to light PA, while the HRave corresponds to light to moderate PA [25]. Consequently, the game used in the present study seemed to generate a low amount of PA, and thus seems to be insufficient to accommodate the WHO guidelines of more exercise at moderate or vigorous intensities in adult populations. On the other hand, the updated guidelines highlight that every move counts, and to limit sedentary behavior [12]. This means that also PA of very light to light intensity may be of importance to accommodate the consequences of sedentary behavior and to create a more physically active adult population. Furthermore, active video and VR games have shown to have higher scores of enjoyment and to be a more motivating form of PA [3,6].”
9. Overall, I like this paper and that you are exploring VR gaming as a means to encourage physical activity. Some minor edits would improve this.

Reviewer 2 Report
Thank you for the opportunity to review this manuscript. The manuscript provided a brief report for understanding the accuracy of different measurements in energy expenditure. I only have a few questions.
1. “Our hypotheses were: 1) wrist-worn accelerometers would estimate EE more accurate than waist-worn accelerometers, and 2) HR measurements would overestimate EE.”
What is the rationale behind these hypotheses? The authors may need to introduce the pros and cons of different measurements or describe the known results from the existing literature.
2. Please also provide a heading for the paragraph of participants, e.g. “ 2.1. Participants”.
3. Regarding the test procedures, is the total time for each session was 40 minutes (10 for warm up + 30 for real measurement)?
4. It would be clearer if authors describe the test procedures and measurement devices separately. That is, for example, “2.2. Test procedures; 2.3. Measurements”
Author Response
First of all, thank you for a thorough review of our manuscript. Please see our point-by-point response to your comments below.
1. “Our hypotheses were: 1) wrist-worn accelerometers would estimate EE more accurate than waist-worn accelerometers, and 2) HR measurements would overestimate EE.”
What is the rationale behind these hypotheses? The authors may need to introduce the pros and cons of different measurements or describe the known results from the existing literature.
Author response:
We have included the following sentences in the introduction in the revised manuscript (Line 49) and we hope that this provides a sufficient rationale behind our hypotheses: “However, well-known limitations of portable devices such as accelerometers and heart rate (HR) monitors question the accuracy of such devices for determining EE during active VR gaming [3,10,15]. For instance, accelerometers attached to the wrist and waist have previously shown large differences in estimations of both sedentary activity and moderate-to-vigorous physical activity (MVPA) while active VR gaming [3]. In that study, waist-worn accelerometers seemed to consequently estimate a lower exercise intensity compared to wrist-worn accelerometers over a variety of active VR games. However, both sites provided similar estimates of exercise intensity during walking. This may indicate an underestimation of EE by waist-worn accelerometers when the upper-body is predominantly used, or an overestimation by wrist-worn accelerometers in active VR games. Perrin et al. [10] criticizes the use of heart rate (HR) to evaluate intensity and EE during active video gaming, and reported a possible overestimation of exercise intensity, and thus EE, by HR-measurements. Pope et al. [15] reported less valid estimations of EE in comparisons of accelerometers and well-known HR based smartwatches, although without comparisons to indirect calorimetry.
2. Please also provide a heading for the paragraph of participants, e.g. “ 2.1. Participants”.
Author response:
We have changed this accordingly (Line 81).
3. Regarding the test procedures, is the total time for each session was 40 minutes (10 for warm up + 30 for real measurement)?
Author response:
This is correct. We have now revised this sentence to (Line 111) “Two to seven days after the VO2max test, a 40 min VR-testing session was performed in the laboratory. This session consisted of a 10-min warm-up followed by 30-minute active VR gaming.”
4. It would be clearer if authors describe the test procedures and measurement devices separately. That is, for example, “2.2. Test procedures; 2.3. Measurements”
Author response:
We have now changed the method-section according to your comment (Line 93).

Reviewer 3 Report
This article has a very interesting and current theme, although the document itself is well structured, the abstract needs to be revised, it is not very "reader-friendly".
The research carried out in this article seems to me to be in line with expectations, the references used are adequate and I believe that the objective of this study brings added value to research to this topic in particular.
However, I feel the conclusion is very vague and small and should be revised, taking into account the last comment below.
Overall, after some minor corrections, it is an interesting article.
Errors/questions/comments:
Line 32: "although types of games [3], and VR equipment [11] affect EE and PA level." - This phrase confuses the reader, please review it.
Line 35: "in 2020 to recommendation for adults" - This phrase confuses the reader, please review it.
Line 37: "Further, the new guidelines highlight that every move counts, and to limit sedentary behavior" - This phrase confuses the reader, please review it.
Line 44 - What is MVPA? I think is not extented/explained in the document.
Line 91: "Characteristics of the different difficulty levels in the game are presented quantitatively in Table 1" - A short description of what is in the table should be given to help the reader better understand the values.
Line 133: "and the level of significance was set to 0.05" - Why?
Line 141: "Physiological and anthropometrical characteristics of the participating subjects are presented in Table 2." - Once again, a short description of what is being presented in the table is welcome.
Line 265: "However, the low number of participants, a homogenous sample and evaluations of several active VR games limit generalization and the present data should be confirmed in larger studies" - The number of participants is really too low to be able to draw definitive conclusions, but in my view the study itself is well prepared and with well-defined objectives.
Line 271: "To conclude, a 30-min active VR gaming session identified that wrist-worn accelerometers generated the most accurate EE estimates compared to indirect calorimetry, while HR and waist-worn accelerometers showed inaccurate estimates. The findings may have methodological implications for population-based surveillance studies of PA levels due to increased participation in active VR gaming." - Attention to this conclusion, in this specific case with this game in particular the result is this, but in another type of game it may not be, mainly one that forces the player to move physically over a small area/zone.
Author Response
First of all, thank you for your thorough review of our manuscript. Please see our response to your comments below.
1. This article has a very interesting and current theme, although the document itself is well structured, the abstract needs to be revised, it is not very "reader-friendly".
Author response:
We have now made changes to the abstract and we hope the changes make the abstract more reader-friendly
2. The research carried out in this article seems to me to be in line with expectations, the references used are adequate and I believe that the objective of this study brings added value to research to this topic in particular.
However, I feel the conclusion is very vague and small and should be revised, taking into account the last comment below.
Overall, after some minor corrections, it is an interesting article.
Errors/questions/comments:
3. Line 32: "although types of games [3], and VR equipment [11] affect EE and PA level." - This phrase confuses the reader, please review it.
Author response:
We have now revised this sentence (Line 33): “EE has been found to be comparable to traditional activities, e.g., walking and running [10]. However, previous studies have reported that different VR games, different game difficulty levels, and VR equipment all have an impact on both EE and PA level [3,10,11].”
4. Line 35: "in 2020 to recommendation for adults" - This phrase confuses the reader, please review it.
Author response:
We have revised this sentence (Line 37): “World Health Organization (WHO) updated their guidelines on PA and sedentary behavior in 2020, and recommend either 150-300 minutes/week of moderate intensity PA, 75-150 min/week of vigorous intensity PA, or a combination of the two for adults [12].
5. Line 37: "Further, the new guidelines highlight that every move counts, and to limit sedentary behavior" - This phrase confuses the reader, please review it.
Author response:
We have now revised this sentence (Line 40): “Further, the updated guidelines highlight that instead of the previous recommendations of a minimum of 10-min bouts of physical activity in order for it to count, all physical activity and every move now counts when accumulating daily activity level. To limit sedentary behavior is now also a separate recommendation from the 2020 guidelines. This means that also PA of very light to light intensity may be of utmost importance to accommodate the consequences of sedentary behavior.”
6. Line 44 - What is MVPA? I think is not extented/explained in the document.
Author response:
MVPA is moderate-to-vigorous intensity physical activity. This is now explained in the revised manuscript (Line 53).
7. Line 91: "Characteristics of the different difficulty levels in the game are presented quantitatively in Table 1" - A short description of what is in the table should be given to help the reader better understand the values.
Author response:
We have now revised this sentence to (Line 158): “An overview of the total amount of cube slashes and wall dodges in the different difficulty levels in the game are presented quantitatively in Table 1.”
We have also revised the table caption for Table 1 to “Values are the total amount of cubes to slash with both hands, the right hand, the left hand, and slashes per second that the player should try to perform at the different difficulty levels. Total amount of walls, upper walls, right walls, and left walls that the player should try to dodge at the different levels of difficulty are also displayed. The values indicate the number of slashes and walls that appear throughout the song each time the same song is played”.
8. Line 133: "and the level of significance was set to 0.05" - Why?
Author response:
The level of significance was set to an alpha value of p < 0.05, and this is now included in the revised manuscript (Line 194). This significance level was chosen based on sample size, study design and significance level of comparable studies.
9. Line 141: "Physiological and anthropometrical characteristics of the participating subjects are presented in Table 2." - Once again, a short description of what is being presented in the table is welcome.
Author response:
We have now revised this sentence (Line 202): “Age, body-weight and physiological results from the VO2max test and the 30 min active VR gaming session (i.e., PA levels and EE from indirect calorimetry, HR and accelerometers) are presented in Table 2.”
10. Line 265: "However, the low number of participants, a homogenous sample and evaluations of several active VR games limit generalization and the present data should be confirmed in larger studies" - The number of participants is really too low to be able to draw definitive conclusions, but in my view the study itself is well prepared and with well-defined objectives.
Author response:
Thank you for your comment. We agree with you, and we have revised this sentence (Line 336): “However, the low number of participants, a homogenous sample of sedentary young adults and lack of evaluations of several active VR games limit generalization and the present data should be interpreted with caution in relation to other populations.”
11. Line 271: "To conclude, a 30-min active VR gaming session identified that wrist-worn accelerometers generated the most accurate EE estimates compared to indirect calorimetry, while HR and waist-worn accelerometers showed inaccurate estimates. The findings may have methodological implications for population-based surveillance studies of PA levels due to increased participation in active VR gaming." - Attention to this conclusion, in this specific case with this game in particular the result is this, but in another type of game it may not be, mainly one that forces the player to move physically over a small area/zone.
Author response:
We do see your point here, and we have now revised the conclusion: “To conclude, a 30-min active VR gaming session identified that wrist-worn accelerometers generated the most accurate EE estimates compared to indirect calorimetry, while HR and waist-worn accelerometers showed inaccurate estimates. The findings may have methodological implications for population-based surveillance studies of PA levels due to increased participation in active VR gaming. However, these conclusions are based on one type of active VR game in a homogenous population of young, sedentary adults. Thus, future studies should investigate these differences in a broader range of active VR games and other populations. For instance, games that forces the player to move physically over a smaller area and incorporation of both upper- and lower body movements is warranted.”

Round 2
Reviewer 2 Report
The authors addressed all my concerns. Although there are some typos, I believe it can be handled by proofread before publishing.